# A Study of Maternal Patients Diagnosed with Inborn Errors of Metabolism Due to Positive Newborn Mass Screening in Their Newborns

**DOI:** 10.3390/children10081341

**Published:** 2023-08-03

**Authors:** Takanori Onuki, Shota Hiroshima, Kentaro Sawano, Nao Shibata, Yohei Ogawa, Keisuke Nagasaki, Hiromi Nyuzuki

**Affiliations:** Division of Pediatrics, Department of Homeostatic Regulation and Development, Niigata University Graduate School of Medicine and Dental Sciences, Niigata 951-8510, Japan; t.onuki7216@gmail.com (T.O.); sho980522@gmail.com (S.H.); sawano@med.niigata-u.ac.jp (K.S.); shibata8400@gmail.com (N.S.); yohei_oga@yahoo.co.jp (Y.O.); nagasaki@med.niigata-u.ac.jp (K.N.)

**Keywords:** newborn screening, maternal inborn errors of metabolism, primary systemic carnitine deficiency, 3-Methylcrotonyl-CoA carboxylase deficiency, false-positive newborn screening

## Abstract

Background: There are reports of mothers being diagnosed with inborn errors of metabolism (IEM) via positive newborn screening (NBS) of their newborns. Mothers with IEM are often considered to have mild cases of little pathological significance. Based in Niigata Prefecture, this study aimed to investigate mothers newly diagnosed with IEM via positive NBS in their newborns using tandem mass spectrometry, and to clarify the disease frequency and severity. Methods: This was a single-institution, population-based, retrospective study. The subjects were mothers whose newborns had false-positive NBS, among 80,410 newborns who underwent NBS between April 2016 and May 2021. Result: there were 3 new mothers were diagnosed with IEM (2 with primary systemic carnitine deficiency (PCD) and 1 with 3-methylcrotonyl-CoA carboxylase deficiency) out of 5 who underwent examination among 18 false positives. The opportunity for diagnosis was low C0 and high C5-OH acylcarnitine levels in their newborn. Two novel *SLC22A5* variants (c.1063T > C/c.1266A > G) were identified in patients with PCD. None of the patients had any complications at the time of diagnosis, but two patients showed improvement in fatigue and headache after taking oral carnitine. Conclusion: New mothers with IEM cannot be considered as mild cases and need to be treated when necessary. The two novel *SLC22A5* variants further expand the variant spectrum of PCD.

## 1. Introduction

Newborn screening (NBS) is performed worldwide. Although the aim of NBS is to identify newborn patients and to improve prognosis through early therapeutic intervention, some reports have shown that mothers of NBS-positive newborns were diagnosed with inborn errors of metabolism (IEM) such as primary systemic carnitine deficiency (PCD, OMIM # 212140) and 3-Methylcrotonyl-CoA carboxylase deficiency (3-MCCD, OMIM # 210200) after positive NBS [1,2,3]. NBS using tandem mass spectrometry (MS/MS) was introduced in Japan in 2014. Screening is currently conducted for 20 diseases, including amino acid metabolism disorders, urea cycle disorders, organic acid metabolism disorders and fatty acid metabolism disorders, as well as galactosemia, congenital hypothyroidism and congenital adrenal hyperplasia [4]. However, the number of newly diagnosed mothers with IEM and the diseases most common in Japan are unknown. Mothers newly diagnosed with IEM are considered to be asymptomatic or mild cases of little pathological significance [1,2,3]. However, some reports have shown that a mother newly diagnosed with IEM had symptoms or was a severe case [5,6]. Therefore, the severity including genotype–phenotype correlation is not known. Moreover, the natural course of IEMs diagnosed into adulthood is unknown. The aim of this study, based in Niigata Prefecture, Japan, was to investigate new mothers newly diagnosed with IEM after positive NBS in their newborns using tandem mass spectrometry, and to elucidate the disease frequency and severity.

## 2. Materials and Methods

This was a single-institution, population-based, retrospective study. A total of 80,410 newborns underwent NBS between April 2016 and May 2021, in Niigata prefecture, Japan.

### 2.1. NBS in Niigata Prefecture

Blood samples were collected on filter paper within the first 4 to 6 postnatal days and then examined using tandem mass spectrometry (Waters^®^ ACQUITY^®^ TQ Detector, Milford, MA, USA). All NBS samples were centralized at the Niigata Health Laboratory Center, and all subjects who were positive for NBS via tandem mass spectrometry were examined in our hospital. Amino acids (phenylalanine, leucine, isoleucine, valine, methionine, and citrulline), acylcarnitine (C2, C3, C5, C5DC, C5OH, C8, C10, C14, C14:1, C16OH, and C18:1) and free carnitine (C0) were measured. The target IEM diseases included the following: Phenylketonuria (PKU), Maple syrup urine disease, Homocystinuria (MSUD), Citrullinemia type 1, Argininosuccinic aciduria, Methylmalonic acidemia, Propionic acidemia, Isovaleric acidemia, 3-MCCD, 3-hydroxy-3-methylglutaric acidemia, Multiple carboxylase deficiency, Glutaric acidemia type 1, Medium-chain acyl CoA dehydrogenase deficiency (MCADD), Very long chain acyl CoA dehydrogenase deficiency, Trifunctional protein deficiency (VLCADD), Carnitine palmitoyl transferase 1/2 deficiency and Galactosemia. In addition, PCD, Carnitine-acylcarnitine translocase deficiency, Beta-ketothiolase deficiency, Citrin deficiency or Glutaric acidemia type 2 can be detected via this method.

### 2.2. Subjects of This Study

We examined the physical findings, blood tests and urinalysis results of NBS-positive newborns. Specifically, urinary organic acids, serum acylcarnitine and serum amino acids were analyzed to detect disorders of organic acid metabolism disorders, fatty acid metabolism disorder and amino acid metabolism disorders, respectively. If the results of these examinations were false positives, we considered that some markers such as acylcarnitine and free carnitine may have maternal influences and recommended that the mother be tested at our hospital. Consent from the mothers was obtained, and physical findings, blood tests and urinalysis (the same examination as that for their newborns) for the mothers were performed. Among the mothers who underwent the examinations, those who were newly diagnosed with IEM were selected in this study. We retrospectively reviewed the opportunities for the diagnosis, definitive diagnosis, symptoms, examination and treatment status of mothers newly diagnosed with IEM.

## 3. Results

Among 80,410 newborns, 26 were NBS positive for IEM. All newborns visited our hospital, and of the 26 newborns that were NBS positive for IEM, 8 were clinically diagnosed with IEM including PKU, MSUD, MCADD, VLCADD, type 4 Galactosemia and Citrin deficiency. There were 18 newborns with an NBS-false-positive diagnosis for IEM, defined as cases that spontaneously improved without abnormalities via close examinations. Of the mothers of the 18 NBS-false-positive newborns, 5 consented to undergo examinations, of which 4 cases had low C0, while 1 case had low C0 and high C5OH. As a results, three mothers were newly diagnosed with IEM (two diagnosed with PCD, one with 3-MCCD). The other two mothers had no abnormalities, and their diagnosis was considered to be due to their newborn’s post-natal nutritional conditions. Newborns whose mothers did not consent to examinations were diagnosed with transient galactosemia, methioninemia and tyrosinemia (Figure 1). The patients included in this study are listed in Table 1.

### 3.1. Patient 1

Patient 1 was aged 30 years and had no symptoms and no problems during the course of her first pregnancy. She visited our hospital due to low C0 level (3.64 μmol/L, cut-off 7.0 μmol/L) in NBS of her newborn. Her newborn had only low C0 levels (4.16 μmol/L) and was treated with oral carnitine (300 mg/day = 100 mg/kg/day). The condition of the newborn improved, with treatment stopped at 11 months. Patient 1 also underwent a blood test, and it revealed markedly low blood free carnitine levels (8.6 μmol/L). Urinalysis did not detect organic acids, but it showed increased carnitine excretion rate (5.4%, normal range: <2.1%). Genetic analysis was performed, which detected heterozygous variants of c. 1063T > C/c. 1266A > G of SLC22A5(NM_003060.4). At diagnosis, the patient had no complications associated with PCD, such as cardiomyopathy. She started taking oral carnitine (2000 mg/day = 37 mg/kg/day), her free carnitine level increased to 24.4 μmol/L, and she experienced improvements in terms of her headache and fatigue symptoms. After treatment, she became pregnant, experienced decreased appetite, suffered from fatigue, and her headache worsened. Although there were no findings of suspected metabolic crisis such as metabolic acidosis or elevated ammonia, as expected, the free carnitine level decreased to 13.0 μmol/L; therefore, we increased the oral carnitine dose (maximum 3250 mg/day = 60 mg/kg/day). Free carnitine level increased to 17.8 μmol/L. We also administered a glucose drip at 15, 19 and 20 weeks of gestation to prevent acute metabolic attack. However, the patient developed a subchorionic hematoma, and abortion was performed because of bleeding and physical strain at 21 weeks of gestation.

### 3.2. Patient 2

Patient 2 was aged 36 years and also has had no symptoms and no problems during the course of her pregnancy. She visited our hospital due to low C0 level (2.71 μmol/L, cut-off 7.0 μmol/L) in NBS of her newborn. Her newborn had only low C0 levels (4.16 μmol/L) and was treated with oral carnitine (300 mg/day = 100 mg/kg/day). The condition of the newborn improved, with treatment stopped at four months. Patient 2 also underwent a blood test and it revealed markedly low blood free carnitine levels (5.9 μmol/L). Urinalysis showed increased carnitine excretion rate (11.8%, normal range: <2.1%). Genetic analysis was performed, which detected heterozygous variants of c.865C > T (rs386134212)/c.1400C > G (rs60376624) of SLC22A5. The patient had no PCD-associated complications at the time of diagnosis. She was started on oral carnitine (2000 mg/day = 31 mg/kg/day), and her free carnitine level increased to 17.8 μmol/L. Before and after taking oral carnitine, her symptoms remained the same.

### 3.3. Patient 3

Patient 3 was aged 42 years and also has had no symptoms and no complications during the course of her pregnancy. She visited our hospital due to low C0 level (4.67 μmol/L, cut-off 7.0 μmol/L) and high C5-OH acylcarnitine (4.38 μmol/L, cut-off 1.0 μmol/L) in NBS of her newborn. Only 3-Methylcrotonylglycine was found in the urine. Her newborn was treated with oral carnitine internally for up to three months. Patient 3 also underwent blood test and it revealed markedly low blood free carnitine levels (4.4 μmol/L) and high C5-OH acylcarnitine level (12.65 μmol/L). Urinalysis detected elevation in the levels of 3-Methylcrotonylglycine and 3-Hydroxyisovaleric acid. Genetic analysis was not performed. She was started on oral carnitine (1200 mg/day = 23 mg/kg/day), and her free carnitine levels increased to 23.3 μmol/L. After treatment of oral carnitine, she experienced improvements in her headache and fatigue symptoms. These improved symptoms might be caused by low free carnitine.

We did not perform genetic analyses for the newborns because close examinations did not reveal any complications, and their condition improved naturally. The diagnosis of Patient 1 and 2 was confirmed via both molecular testing and genetic analysis. On the other hand, the diagnosis of patient 3 was confirmed via molecular testing alone (including urinalysis of 3-hydroxyisovaleric acid and 3-methylcrotonylglycine and low blood free carnitine levels and high C5-OH acylcarnitine levels) because she did not provide consent for genetic analysis. The diagnosis of 3-MCCD was confirmed via molecular testing such as urinary excretion of 3-hydroxyisovaleric acid and 3-methylcrotonylglycine. We also think that it is important to note that the symptoms improved after taking oral carnitine.

## 4. Discussion

Some reports discussed the cases of mothers with IEM diagnosed with positive NBS in their newborns. In a 9-year cohort study of NBS in Denmark, the Faroe Islands, and Greenland, maternal IEM was evident in twelve of 504,049 cases (8 cases of PCD and 4 cases of 3-MCCD) [1]. In other cohort studies, from 364,545 newborns who underwent NBS, five new mothers were diagnosed with PCD in Quanzhou, China [2], and from 236,368 newborns who underwent NBS, six new mothers were diagnosed with PCD in Quanzhou Area, China [3]. In addition to PCD and 3-MCCD, there have been some case reports of new mothers diagnosed with the following IEM after positive NBS in their newborns: glutaric acidemia type I [7], holocarboxylase synthetase deficiency [8], maternal Middle-chain acyl-CoA dehydrogenase deficiency [9,10], very long chain acyl-CoA dehydrogenase deficiency [11], combined homocystinuria and methylmalonic aciduria [12] and multiple acyl-CoA dehydrogenation deficiency due to a riboflavin transporter gene defect [13]. The diagnosis was based primarily on decreased free carnitine levels in newborns and elevated specific acylcarnitine levels in some. Our findings regarding new mothers diagnosed with PCD and 3-MCCD were consistent with the findings of these reports.

Lund et al. [1] reported that all mothers with 3-MCCD and PCD were asymptomatic before diagnosis. Similarly, Lin et al. [2] and Zhou et al. [3] reported that almost all mothers with PCD were asymptomatic. In contrast, there is one report of a new mother with PCD who had dilated cardiomyopathy at diagnosis, and her cardiac function improved after treatment with carnitine supplementation [5]. De Biase I et al. [6] reported a case of a female in her twenties who experienced a syncopal episode caused by ventricular tachycardia, had a prolonged QT interval, became pregnant, and was diagnosed with PCD via a positive NBS in her newborn. Moreover, after starting treatment with carnitine supplementation, no further syncopal episodes occurred, and the QT interval returned to normal. Therefore, asymptomatic maternal IEM patients may not be completely considered as mild cases. Although the causal relationship is unknown, patient 1 had an exacerbation of symptoms and subchorionic hematoma during pregnancy with her second child, even though her treatment had started. She may have needed to increase oral carnitine dosage, prior to pregnancy.

The heterozygous *SLC22A5* genotype of patient 2 (c.865C > T/c.1400C > G) was previously reported. Heterozygous genotypes involving c.865C > T, c.760C > T and c.1400C > G variants were also reported to have decreased activity in fibroblasts derived from an affected patient [14]. We speculate that the variant of c.865C > T is classified as pathogenic according to the American College of Medical Genetics (ACMG) criteria [15], as it meets the classification criteria of PVS1, PS1, PS3, PM2 and PP4; the variant of c.1400C > G is also classified as pathogenic, as it meets the classification criteria of PS1, PS3, PM2 and PP4. On the other hand, the *SLC22A5* variant of patient 1 (c.1063T > C/c.1266A > G) has not been reported. Through in silico analysis, the variant of c.1063T > C was classified as deleterious (prediction score 0.01) via the Sorting Intolerant from Tolerant (SIFT, https://sift.bii.a-star.edu.sg/ (accessed on 10 December 2022)) software and as possibly damaging (prediction score 0.824) via Poly-Phen (http://genetics.bwh.harvard.edu/pph2/ (accessed on 10 December 2022)). Mutation Taster (https://www.mutationtaster.org/ (accessed on 14 February 2022)) revealed the variant of c.1266A > G disease caused by splice site changes. The variant of c.1063T > C meets only PM2, PP3 and PP4 criterion in ACMG guidelines and was classified as a variant of uncertain significance (VUS), and the variant of c.1266A > G, which meets PVS1, PM2, PP3 and PP4 criteria in ACMG guidelines, was classified as likely pathogenic.

There have been some reports of genotype–phenotype correlations in PCD. Rose et al. [16] indicated that the frequency of nonsense mutations is significantly increased in patients presenting symptomatically compared to that in asymptomatic females. For example, a mother with PCD who had c.51C > G/c.760C > T and had experienced a few episodes of syncope since 13 years of age died suddenly one year after delivering a baby [17]. However, there was a report that a mother with PCD, who had homozygous c.760C > T mutation for a few years, had no symptoms to dates [17]; another report stated that one mother with PCD who had cardiomyopathy had a heterozygous of the c.1400C > G/ c.[1400C > G, c.845G > A] mutation [5]. Rasmussen et al. [18] showed that the c.95A > G homozygous genotype for a determinant variant must be considered severe and reported that severe symptoms such as cardiac arrest relating to PCD patients with the c.95A > G homozygous genotype for a determinant variant. A mother with PCD with a syncopal prolonged QT interval had c.95 A > G and c.136 C > T [14]. In the present study, our patient 2 had no specific symptoms before diagnosis; however, since patient 2 had nonsense mutations that were identified, the patient should therefore not be classified as a mild case based on these reports. However, because there is no distinct genotype–phenotype correlation, we believe that maternal IEM patients should be carefully monitored and treated.

The opportunity for the diagnosis of the mothers in our study is mainly low C0 level in NBS. In non-vegetarians, most carnitine sources (~75%) are obtained from the diet, whereas endogenous synthesis accounts for approximately 25%. Renal carnitine reabsorption, dietary intake and endogenous production maintain carnitine homeostasis [19]. Although an average carnitine level in strict vegetarians (vegans) is markedly reduced compared with that in non-vegetarians, because of endogenous synthesis and renal carnitine reabsorption, average carnitine level is 32.3 ± 1.5 µmol/L [20]. Because our cut-off level of C0 was 7.0 micro mol/L, factors other than diet may be involved in causing such low levels. Therefore, mothers of newborns with low C0 levels should be examined closely.

This study had some limitations. This was a single-institution, retrospective study with a small number of cases. Moreover, only 5 mothers underwent examinations from the 18 mothers with NBS-false-positive newborns. However, thirteen newborns, whose mothers did not undergo examinations, were diagnosed with transient galactosemia, methioninemia and tyrosinemia; hence, we believe that these conditions are less likely to be influenced by maternal factors.

## 5. Conclusions

Three new mothers (two with PCD and one with 3-MCCD) were newly diagnosed with IEM via NBS positivity among 80,140 newborns who underwent NBS in Niigata Prefecture, Japan, between April 2016 and May 2021. The opportunity for diagnosis was low C0 and high C5-OH acylcarnitine levels in their newborns. New mothers with IEM cannot be considered as mild cases and should be carefully monitored and treated when necessary. The two novel *SLC22A5* variants further expanded the variant spectrum of PCD.

## Figures and Tables

**Figure 1 children-10-01341-f001:**
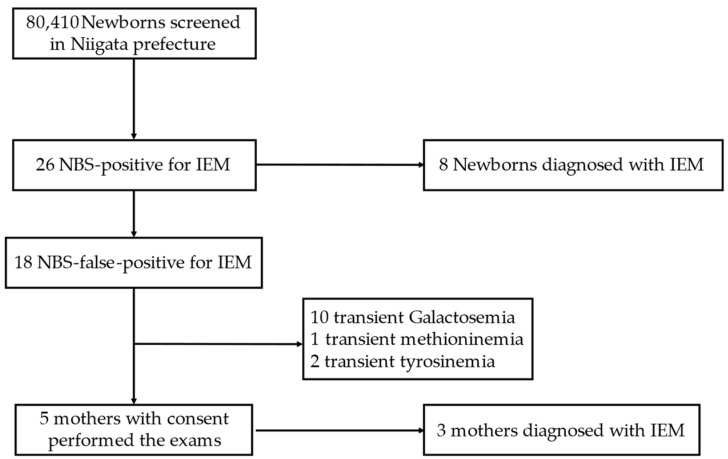
Enrollment of the study subjects. IEM, inborn error of metabolism; NBS, newborn screening.

**Table 1 children-10-01341-t001:** New mothers diagnosed with inborn errors of metabolism via positive NBS in their newborn. PCD, primary systemic carnitine deficiency; 3-MCCD, 3-Methylcrotonyl-CoA carboxylase deficiency; y, years; NBS, newborn screening.

	Patient 1	Patient 2	Patient 3
Diagnosis	PCD	PCD	3-MCCD
Age(y)	30	36	42
Opportunity of diagnosis	Low C0 of NBS	Low C0 of NBS	Low C0 and high C5-OH of NBS
Clinical presentation	No symptoms	No symptoms	No symptoms
Examination	Free carnitine 8.6 μmol/L in blood	Free carnitine 5.9 μmol/L in blood	Free carnitine 4.4 μmol/L in blood 3-Methylcrotonylglycine and 3-Hydroxyisovaleric acid Elevation in urine
Gene analysis	*SLC22A5* c.1063T > C/ c.1266A > G	*SLC22A5* c.865C > T/ c.1400C > G	Not implemented
Diagnosis of newborn	Secondary to low maternal plasma carnitine levels	Secondary to low maternal plasma carnitine levels	Secondary to low maternal plasma carnitine levels
Treatments	Oral carnitine	Oral carnitine	Oral carnitine
Symptoms after treatment	Improvement in headache and fatigue	No change	Improvement in headache and fatigue

## Data Availability

The data presented are available upon request from the corresponding author. The data are not publicly available because of privacy restrictions.

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
