# Peer review of "A Study of Maternal Patients Diagnosed with Inborn Errors of Metabolism Due to Positive Newborn Mass Screening in Their Newborns"

_children, 2023, doi:10.3390/children10081341_

Round 1
Reviewer 1 Report
Dear colleagues, thank you for the effort in writing this article.
However, I believe that the cases are not numerous enough to have significant results. The 3 diagnoses that are made in the mothers: two PCD and one 3-MCCD, refer to pathologies commonly found in the mothers of the patients who are diagnosed. On the other hand, the study of the parents following each newborn screening diagnosis is a custom now performed everywhere. Finally, the symptomatology of patient 3, which has improved following the administration of carnitine, does not seem characteristic of this organic aciduria, but more characteristic of PCD.
Author Response
Point 1: Dear colleagues, thank you for the effort in writing this article.
However, I believe that the cases are not numerous enough to have significant results. The 3 diagnoses that are made in the mothers: two PCD and one 3-MCCD, refer to pathologies commonly found in the mothers of the patients who are diagnosed. On the other hand, the study of the parents following each newborn screening diagnosis is a custom now performed everywhere. Finally, the symptomatology of patient 3, which has improved following the administration of carnitine, does not seem characteristic of this organic aciduria, but more characteristic of PCD.
Response 1:
We appreciate for reviewer’s comment.
We have added limitations of this study as single-institution, retrospective study and the small number of cases at line243-248. Because we also think that improved symptoms of patient 3 might be caused by low free carnitine, we have added it at line 151-152. (in red)
Inborn errors of metabolism are inherited diseases and have regional characteristics. Because there are similar reports in other regions, but few reports in our region, we believe our study is worth reporting.
Reviewer 2 Report
It is retrospective study to investigate the possibility of diagnosing mild metabolic diseases in mothers of children with false positiv NBS results. The results are presented in a short report.
The data are potential interesting and confirm earlier findings. The English and the style of the writing should be improved.
Additional comments.
1. Please provide a reference for lines 44-46
2. Why did the authors not perform genetic analyses in the babies ?
3. Please describe the weaknesses of the study in your paper. Can you really be sure of the diagnosis in the mothers?
English style and wording should be improved by native or near native speaker
Author Response
Point 1: It is retrospective study to investigate the possibility of diagnosing mild metabolic diseases in mothers of children with false positiv NBS results. The results are presented in a short report.
The data are potential interesting and confirm earlier findings. The English and the style of the writing should be improved.
Response 1:
We appreciate for reviewer’s comment.
English text has been recalibrated and corrected. (in red)
Point 2: Additional comments.
- Please provide a reference for lines 44-46
Response 2-1:
We have provided reference at line 44-46. The reference numbers have been changed accordingly. (in red)
- Why did the authors not perform genetic analyses in the babies ?
Response 2-2:
We did not perform genetic analyses for the newborns, because close examinations did not reveal any complications and their condition improved naturally. We have added it in mainscript at line 153-154. (in red)
- Please describe the weaknesses of the study in your paper. Can you really be sure of the diagnosis in the mothers?
Response 2-3:
We have added limitations of this study as single-institution, retrospective study and the small number of cases at line 243-248. We have also mentioned about the certainty of the diagnosis in the mothers at line 155-162. (in red)
Point 3: Comments on the Quality of English Language
English style and wording should be improved by native or near native speaker
Response 3:
English text has been recalibrated and corrected.(in red)
Reviewer 3 Report
The investigators leveraged newborn screening for IEM to access the potential clinical status of the birth mothers. Among the false-positive newborns, five women consented and three were identified with IEM. Then the authors thus provide a case series of mothers diagnosed for the first time after a false-positive newborn result. New novel variants of SLC22A5 were identified and reported. The study extends findings in the literature to justify maternal diagnostic investigation when a newborn is false-positive for IEM.
My copy edit recommendations in brackets [].
Page 1. Abstract,
Authors need to indicate the total number of maternal consents to give a perspective of the cohort, that out of 18 false-positive pregnancies, 5 had consented and 3 were considered cases.
Such as: Line #21 after "were diagnosed" [out of 5 who consented among 18 false-positives].
Or something like the above.
#25 New mother[s]
#26 “and need to [be] treat[ed] when necessary.”
Pages 5-6.
Discussion.
I had some comments here but upon reexamination I found that the authors did make a solid case that maternal patients with a false-positive newborn should be clinically examined and should not be considered mild cases prior to examination. I think the emphasis is appropriate.
Limitations: I suggest that the authors mention briefly in the discussion that the results of the study were limited by the small number of maternal subjects (5) who agreed to be examined out of 18 total false positives in the study cohort. So, the high proportion or expectation of 3 new maternal IEM cases out of 5 examined cannot be generalized to other mothers with a false positives newborn.
This limitation does not refute the assertions made by the authors about this case series. The remainder of the 18 women may have no metabolic problems; the ones who consented may have done so due to their own sub-clinical health issues, but we don’t know.
This is a well-written manuscript needing minor and forgivable copy edits.
My minor copy edits have already been mentioned for the abstract.
Author Response
The investigators leveraged newborn screening for IEM to access the potential clinical status of the birth mothers. Among the false-positive newborns, five women consented and three were identified with IEM. Then the authors thus provide a case series of mothers diagnosed for the first time after a false-positive newborn result. New novel variants of SLC22A5 were identified and reported. The study extends findings in the literature to justify maternal diagnostic investigation when a newborn is false-positive for IEM.
Point 1: My copy edit recommendations in brackets [].
Page 1. Abstract,
Authors need to indicate the total number of maternal consents to give a perspective of the cohort, that out of 18 false-positive pregnancies, 5 had consented and 3 were considered cases.
Such as: Line #21 after "were diagnosed" [out of 5 who consented among 18 false-positives].
Or something like the above.
#25 New mother[s]
#26 “and need to [be] treat[ed] when necessary.”
Response 1:
We appreciate for reviewer’s comment.
We have corrected as your recommendations. However, we have changed the term to "were performed examinations" rather than "consented". It includes the fact that the mothers did not perform examinations because thirteen newborns whose mothers were not performed examinations were diagnosed with transient galactosemia, transient methioninemia and transient tyrosinemia that are less likely to be maternal influences. It was difficult to understand, so we have also added the methods at line 78-79. (in red)
Point 2: Pages 5-6.
Discussion.
I had some comments here but upon reexamination I found that the authors did make a solid case that maternal patients with a false-positive newborn should be clinically examined and should not be considered mild cases prior to examination. I think the emphasis is appropriate.
Limitations: I suggest that the authors mention briefly in the discussion that the results of the study were limited by the small number of maternal subjects (5) who agreed to be examined out of 18 total false positives in the study cohort. So, the high proportion or expectation of 3 new maternal IEM cases out of 5 examined cannot be generalized to other mothers with a false positives newborn.
This limitation does not refute the assertions made by the authors about this case series. The remainder of the 18 women may have no metabolic problems; the ones who consented may have done so due to their own sub-clinical health issues, but we don’t know.
Response 2:
We have added limitations of this study as single-institution, retrospective study and the small number of cases at line 243-248.
As for the 13 who were not performed examinations, please see above. (in red)
Comments on the Quality of English Language
This is a well-written manuscript needing minor and forgivable copy edits.
My minor copy edits have already been mentioned for the abstract.